# Sequential and Binomial Sampling Plans to Estimate *Thrips tabaci* Population Density on Onion

**DOI:** 10.3390/insects12040331

**Published:** 2021-04-08

**Authors:** Lauro Soto-Rojas, Esteban Rodríguez-Leyva, Néstor Bautista-Martínez, Isabel Ruíz-Galván, Daniel García-Palacios

**Affiliations:** Colegio de Postgraduados, Posgrado en Fitosanidad-Entomología y Acarología, Montecillo, Texcoco 56230, Estado de Mexico, Mexico; rojo@colpos.mx (L.S.-R.); nestor@colpos.mx (N.B.-M.); ruizg.isabel@gmail.com (I.R.-G.); garcia.daniel@colpos.mx (D.G.-P.)

**Keywords:** integrated pest management, spatial distribution, sampling methods, Taylor’s Power Law

## Abstract

**Simple Summary:**

Thrips are tiny insects that cause significant damage to onion crops worldwide. They feed on the plants and can also transmit plant viral diseases. To prevent damage, it is necessary to estimate the population density (average number of insects per plant), through periodic sampling, and to apply a combination of control tactics to maintain thrips at acceptable levels. Conventional sampling methods are precise but require large investments of time and effort. In this study, binomial and sequential sampling plans were developed to estimate thrips population density in a precise and less time-consuming manner. More than 50 onion plots were sampled, and *Thrips tabaci* Lindeman was identified as the predominant pest species. The sampling plans reached acceptable levels of precision (*D* = 0.25) in less time than conventional sampling. Binomial and sequential sampling plans were reliable and easily implemented in practice, but sequential sampling showed better performance than binomial sampling under different field conditions. These findings may help to reduce time and work for *T. tabaci* sampling and, consequently, improve implementation of crop protection tactics on onion.

**Abstract:**

*Thrips tabaci* Lindeman is a worldwide onion pest that causes economic losses of 10–60%, depending on many factors. Population sampling is essential for applying control tactics and preventing damage by the insect. Conventional sampling methods are criticized as time consuming, while fixed-precision binomial and sequential sampling plans may allow reliable estimations with a more efficient use of time. The aim of this work was to develop binomial and sequential sampling for fast reliable estimation of *T. tabaci* density on an onion. Forty-one commercial 1.0-ha onion plots were sampled (sample size *n* = 200) to characterize the spatial distribution of *T. tabaci* using Taylor’s power law (*a* = 2.586 and *b* = 1.511). Binomial and sequential enumerative sampling plans were then developed with precision levels of 0.10, 0.15 and 0.25. Sampling plans were validated with bootstrap simulations (1000 samples) using 10 independent data sets. Bootstrap simulation indicated that precision was satisfactory for all repetitions of the sequential sampling plan, while binomial sampling met the fixed precision in 80% of cases. Both methods reduced sampling time by around 80% relative to conventional sampling. These precise and less time-consuming sampling methods can contribute to implementation of control tactics within the integrated pest management approach.

## 1. Introduction

Onion (*Allium cepa* L.) is one of the most economically important vegetables in the world. It is grown in more than 120 countries with a production that exceeds 100 million tons [1,2]. Although several insects feed on the plant, *Thrips tabaci* Lindeman (Thysanoptera: Thripidae) is considered one of the most damaging pests of onions [3,4]. It reduces the quantity and quality of onions, causing economic losses of 10–60%, depending on the variety, population size, plant phenology, environmental conditions and management [5,6,7,8]. Its importance is attributed to multivoltinism, a high reproduction rate [9,10] and its role as a vector of plant viruses [11,12]. In addition, *T. tabaci* has evolved resistance to several insecticides [13].

Sampling to determine population density is essential for applying control tactics within integrated pest management (IPM) [14,15]. Population density is an indicator of abundance per unit of living space, or per unit of area [16]. In the study of insect pests, the indicator should be estimated using low-cost, representative and precise sampling methods [15,17]. These attributes depend on the sample size and the way in which sampling units are selected. If sampling unit selection is arbitrary and sample size is defined by the researcher, the estimates derived from the sampling have non-calculable precision [18]. Additionally, for developing reliable sampling methods, it is essential to determine the spatial distribution index of the target population [19,20]. Shelton et al. [21] and Fournier et al. [22] reported that *T. tabaci* has a spatial distribution in aggregates. This information is useful for implementing simplified procedures such as binomial or sequential enumerative sampling, which can reduce sampling time and may optimize the sample size using a fixed level of precision [23,24,25].

Binomial sampling, which analyzes the proportion of infested plants and relates it functionally to pest population density, is a useful technique in the evaluation of populations with a high aggregation index, and its implementation reduces sampling time by around 85% [14,26,27]. On the other hand, sequential enumerative sampling, based on counting individuals, optimizes the sample size and depends on pest population density [14,23]. It has been used in the study of arthropod populations [25,28], reducing sampling time by 35–50%, relative to probability sampling methods such as simple random sampling [29]. The latter one includes planning a sampling frame, obtaining a pre-sample, and then estimating the sample size to achieve the desired precision. The objective of this study was to develop and validate binomial and sequential enumerative sampling for reliable and less time-consuming estimation of *T. tabaci* density on onion, which could accelerate decision-making in the IPM strategy.

## 2. Materials and Methods

### 2.1. Sampling of Thrips Populations

Sampling was conducted in commercial onion crops in three states in central Mexico (Puebla, Michoacán, Estado de México), between 18°46′ and 19°59′ N and 97°51′ and 101°14′ W (Figure 1). Sterling (Seminis^®^) (St. Louis, MO, USA) and Carta Blanca (Nunhems^®^) (Leverkusen, Germany) were the predominant onion varieties. Crops were grown using conventional agronomic practices without insecticides. Sampled plants were in the main shoot development, with six or more clearly visible leaves and bulbs around 30% of the expected diameter. Data were obtained from 1.0-ha onion plots. In each plot, systematic sampling was carried out (sample size *n* = 200). Plants were selected beginning at a random starting point at a fixed interval. The total number of thrips was counted on each plant (eggs were not included). Sampling was done by three entomologists; all used a magnifying glass 10× and standardized their criteria by counting thrips on 10 plants and comparing their total numbers. The exercise was repeated five times. From 2015 to 2018, data from 41 different plots were collected and used to develop the sampling plans. Validation was carried out with a different data set from 10 plots sampled from 2019 to 2020. Detailed information on each sampled site was included in the attached Appendix A. In each plot, thrips specimens were collected and preserved in 70% ethanol. Preserved specimens were identified with the aid of available literature [30,31].

### 2.2. Sequential Enumerative Sampling

The sequential sampling method and the optimal sample size were obtained using the Green method [23]. Precision *D* was set at 0.10, 0.15 and 0.25 to limit the standard error as a proportion of the mean. Southwood [19] indicated that with *D* = 0.25, sampling plans with acceptable precision were obtained in different population studies of arthropods. The method used required estimation of parameters *a* and *b* of Taylor’s Power Law [32]. This law describes the relationship between the variance (*s*^2^) and the mean (*m*) through the power equation *s*^2^
*= am^b^*. Parameters were calculated by fitting a linear regression model. Sampling interruption or stop lines and the relationship between the optimal sample size and the mean (*m*) were drawn. To achieve this, Equations (1) and (2), respectively, were used:(1)$$\ln T_{n}=\frac{\ln\left(\frac{D^{2}}{a}\right)}{b-2}+\frac{b-1}{b-2}\times \ln n$$
(2)$$n=\frac{am^{b-2}}{D^{2}}$$
where $T_{n}$ is the accumulated number of insects, *a* and *b* are the Taylor parameters, *D* is the fixed level of precision and *n* is the sample size. Sampling stops if $T_{n}$ crosses the stop line at the chosen precision level. A high frequency of non-infested plants keeps $T_{n}$ away from the stop lines; that is, when *m* approaches zero, the sample size *n* tends toward ∞.

### 2.3. Binomial Sampling

The binomial sampling plan was developed by confirming a functional relationship between the proportion of occupied sampling units (*p*) and the average number of individuals per sampling unit (*m*). This relationship was modeled by the negative binomial function (NB) and parameter *k* was considered a function of the mean (*m*), according to the equation proposed by Wilson and Room [33].
(3)$$p^{\prime }=1-e^{-m\frac{\ln\left(am^{b-1}\right)}{am^{b-1}-1}}$$

Equation (3) was subjected to an iteration process to improve the fit, and the Taylor parameters (*a* and *b*) were proposed as initial values in the iterative process, using the nls2 library of the R program [34,35]. To choose the best model (adjusted NB or NB), a linear regression was performed between the proportion of estimated infested plants (*p′*) and the real values (*p*). The sample size *n* was calculated as a function of the probability of finding non-infested plants $p_{0}$, parameter *k* of the NB and precision *D* = 0.25. To this end, Equation (4) was used [24].
(4)$$n=\frac{1}{D^{2}}\left(1-p_{0}\right)p_{0}{}^{-\left(\frac{2}{k}\right)-1}{\left[k\left(p_{0}^{-\frac{1}{k}}-1\right)\right]}^{-2}$$

### 2.4. Validation of Sampling Plans

Before validation, a general mixed model was used to determine whether there was any effect of onion variety, plant phenology or sampling season on thrips density or spatial distribution. The sampling plans were validated using the analysis of precision obtained (*D′*) by 1000 repetitions of bootstrap resampling [36]. Data from 10 validation plots were processed using the modelr library in the R program [34,37]. For binomial sampling, in each repetition the sample size *n* was calculated based on a pre-sample of 30 plants.

## 3. Results

*Thrips tabaci* accounted for 95% of the individuals collected in the onion plots. Systematic sampling (*n* = 200) produced estimates with a standard error of less than a proportion of 0.10 with respect to the mean *m* (Table 1).

Thrips population density was not influenced by onion variety (F_1,47_ = 2.229, *p* = 0.142), plant phenology (F_1,47_ = 0.163, *p* = 0.688) or sampling season (F_1,47_ = 0.261, *p* = 0.612). Nor was spatial distribution, using the variance/mean ratio, significantly affected by variety (F_1,47_ = 2.278, *p* = 0.138), plant phenology (F_1,47_ = 0.218, *p* = 0.643) or sampling season (F_1,47_ = 0.412, *p* = 0.524). Taylor parameters were obtained using the linear regression model: $\log s^{2}=0.4126+1.511\times \log m$ (F_1,39_ = 1612, *p* < 0.0001) with a setting of R^2^ = 0.9758. The estimated values (*a* = 2.586 and *b* = 1.511) indicated that the *T. tabaci* population had a spatial arrangement in aggregates.

### 3.1. Sequential Enumerative Sampling

The optimal sample size *n* was represented as a function of the mean *m* (Figure 2a). In general, it was observed that, at low *T. tabaci* densities (<2 individuals per plant), a sample size of at least 30 plants was required to obtain estimates with precision *D* = 0.25; for precision *D* = 0.15, it was necessary to evaluate more than 80 plants. Finally, precision *D* = 0.10 required a sample size larger than 180. In the case of more severe infestation of *T. tabaci* (for example, a population density ≥ 30 individuals per plant), a sample size of *n* = 50 would be sufficient to achieve the highest precision established in this study (*D* = 0.10). The onion plots had different population densities (0.20–27.44 individuals per plant); for infestations of more than 40 thrips per plant, sample sizes of less than 45 would be required for the three levels of established precision.

For *D* = 0.25, when the accumulated $T_{n}$ reached 100 individuals, with *n* = 20, sampling was interrupted and the population density was estimated with *m* = $T_{n}/n$ = 100/20 = 5 (Figure 2b); this was confirmed using the figure of optimal sample size, where an average density of *m* = 5 could be estimated by examining approximately 20 sampling units (Figure 2a).

The efficacy of sequential enumerative sampling to reach the fixed precision (*D*) was verified. For example, in plot one, a density of *m* = 0.37 individuals per plant was recorded; sequential sampling derived an average estimate of *m’* = 0.37. This sampling plan was repeated 1000 times and stopped whenever $T_{n}$ crossed the stop line. For *D* = 0.10, the sample sizes ranged between 285 and 797; the obtained precision *D’* = 0.086 ± 0.002 met the pre-established requirements (Table 1). Similarly, in plot five, a density of *m* = 5.82 was recorded, and sequential sampling with *D* = 0.25 estimated an average of *m’* = 5.80. Repetitions of the process produced sample sizes that varied between 14 and 23; the precision obtained *D’* = 0.191 ± 0.004 indicated that the sampling plan was reliable and satisfactory (*D’* ≤ *D*) in more than 95% of the repetitions performed. The samplings in which *D’* > *D* occurred with greater probability in plots with density lower than two individuals per plant and was more frequent if the most demanding precision was set (*D* = 0.10).

### 3.2. Binomial Sampling

Binomial sampling required fitting data to the NB function to relate the proportion of infested plants *p* and the average density *m*. The iterative process showed that the values of *a* = 3.26 and *b* = 1.70 produced a better fit of the NB function (Figure 3a). The regression between the proportion of infested plants estimated with NB (*p′*) and the proportion of infested plants registered in the samplings (*p*) showed an adjustment of R^2^ = 0.9594 (F_1,39_ = 948, *p* < 0.0001); the indicator increased with the NB adjusted by iteration R^2^ = 0.9656 (F_1,39_ = 1125, *p* < 0.0001). The differences between models seem negligible, but a proportion *p* = 0.80 was related to *m′* = 4 for the NB and *m′* = 8 for the adjusted NB; these differences increased with *p* > 0.8 (Figure 3a).

The sample size was a function of the mean *m* (Figure 3b). It was estimated that for low densities of the pest (<1 individual per plant) more than 80 plants were required for estimates with precision *D* = 0.25, while for densities between 2 and 8 thrips per plant, it was sufficient to examine 80 plants to achieve the predetermined precision. As population density increased, it was necessary to increase the sample size to achieve the desired precision.

The precision obtained was satisfactory (*D′* ≤ *D*) in eight of 10 validation plots. Samples in which *D′* > *D* occurred under conditions of low density (<0.5 individuals per plant) or when the proportion of infested plants was very high, relative to the population density.

In most cases, binomial sampling reached the precision level *D* = 0.25. In plot one, a density of *m* = 0.37 individuals per plant was recorded. The sampling plan, through the adjusted NB, derived an average estimate of *m’* = 0.28. In each repetition the sample size *n* was determined by the $p_{0}$ of the pre-sample, so that *n* varied between 90 and 366. The precision obtained, *D′* = 0.31 ± 0.005, did not meet the pre-established requirements (Table 2). Similarly, in plot seven a density of *m* = 9.82 was registered; binomial sampling estimated an average of *m′* = 12.31, repetitions of the process produced sample sizes that varied between 118 and 1023. The precision obtained, *D′* = 0.26 ± 0.009, did not achieve the accuracy of the sampling plan.

## 4. Discussion

Taylor parameters were obtained by combining data from the first years of sampling 41 onion plots. *T. tabaci* showed a spatial distribution in aggregates b = 1.511. Earlier studies of *T. tabaci* on onion crops had previously described this behavior [4,22]. Jiménez et al. [38] and Quiñones et al. [39] found the same distribution for different species of thrips on potato and gladiolus. Sardana et al. [40] indicated that aggregate distribution of *Thrips palmi* Karny on cucurbits could be explained, in part, by oviposition behavior since females preferred to lay eggs in some section of the plant tissue. Other authors, such as Sedaratian et al. [41], attributed the distribution of *T. tabaci* in aggregates to the parthenogenetic reproduction of the species.

Our sampling plans were validated using data sets from 10 independent plots in the same region, following recommendations of different authors [29,42,43], and bootstrap resampling techniques were used to check the precision obtained under various conditions, as suggested by Naranjo and Hutchison [44].

Green’s [23] sequential sampling produces estimates with fixed precision. The method does not require an exhaustive sampling frame; therefore, sampling time is notably reduced, and data collection is interrupted when the pre-established precision is reached, near the population mean [42]. In our study, the systematic samplings performed in 41 onion plots, each with *n* = 200, took between 3 and 4 h. In contrast, if population density was greater than or equal to five thrips per plant (Figure 2a, *D* = 0.25), the sequential enumerative sampling required a lower *n*, and sampling time was reduced to around 30 min per plot. The sample size depends on the desired precision, which is in function of the sampling objectives [19]. For population density estimates, such as those required in the IPM strategy, sequential sampling optimizes effort and requires a moderate number of sampling units, sufficient to meet the expected precision. Under these conditions, with *D* = 0.25, an acceptable level of precision is achieved [19,25,45]. In this way, when population density was greater than or equal to 2.5 thrips per plant, a sample size no greater than 30 sampling units was needed. In contrast, to estimate the same density using precision *D* = 0.10, a maximum of 180 plants would need to be examined (Figure 2a).

It was generally observed that at low densities (<2 thrips per plant) more than 80 sampling units were needed for precision estimates with *D* = 0.15. Cabrera et al. [46] reported similar results in sampling *T. palmi* in potato. They indicated that more than 90 sampling units were needed for sequential sampling of that pest. In the case of low population densities, as *m* approaches zero, *n* increases to impractical levels, undermining the advantages of sequential sampling. For populations with these characteristics, in the context of IPM, it would be enough to know that the pest population density is below the economic threshold, and a control tactic is not needed.

The representation of the stop lines (Figure 2b) on printed data record sheets is useful for implementing a sequential sampling plan. However, time optimization could also be achieved with a dynamic graph on a mobile device. The variability in sample sizes (Table 1) occurred because a different random sample was taken in each repetition. For example, in plot three with *D* = 0.25 the maximum sample size was 51 and the minimum was 23, in the first case a high frequency of non-infested plants was obtained by chance, so the accumulated $T_{n}$ slowly increased.

For implementation of a binomial sampling plan, estimation of population density required calculating the proportion of infested plants with a previous sampling. Once this proportion was obtained, it was possible to infer the preliminary density from the adjusted NB function (Figure 3a). Sampling was then continued until the suggested sample size was met (Figure 3b), and pest density was again inferred. In this way, sampling time was substantially reduced since only plants were checked to verify the presence or absence of a pest. Similarly, Lindenmayer et al. [47] evaluated the aphid *Melanaphis sacchari* Zehntner on sorghum. Their results indicated that binomial sampling was a precise, fast and easy technique to implement in the field. Binns and Bostanian [48] proposed a method to define a cut-off *T* value, a number above which the plant is considered infested, while at equal or lower values it would not. González et al. [25] indicated that choosing a high *T* assumed an increase in effort, as individuals had to be counted to decide whether the plant was infested or not. In our study, for sampling *T. tabaci* on an onion, a cut-off value *T* = 0 was used. The binomial sampling plan reduced evaluation time by more than 80%, relative to conventional systematic sampling and provided estimates with the established precision (*D* = 0.25) in 80% of the cases. Another advantage of this sampling plan is that it does not require highly trained personnel since evaluation is reduced to recording the presence or absence of thrips, unlike count-based methods in which the result depends on the ability of the evaluator to detect and register individuals [49]. However, certain conditions can affect the results of binomial sampling [25,50]. With databases in which *D′* > *D*, it is possible that the pest was initiating colonization (plot one) or there was a constant entry of new individuals (plot seven) because of dispersal from nearby plots. It is also possible that environmental conditions or crop management influenced the spatial distribution of the pest, as reported in other sampling assays [40]. Under these conditions, aggregation indices tend to be low and affect the results.

## 5. Conclusions

Binomial (*D* = 0.25) and sequential enumerative (*D* = 0.10, 0.15 and 0.25) sampling plans were reliable in estimating the population density of *T. tabaci* at the proposed precision levels. In the case of *T. tabaci* on an onion, sequential enumerative sampling allowed a precise and rapid estimate of the pest population density, reducing sampling time and invested effort.

Sequential enumerative sampling showed better performance under different field conditions. Fixed precision levels were achieved in plots with several population densities. The precision of binomial sampling could be affected by the pest aggregation index.

## Figures and Tables

**Figure 1 insects-12-00331-f001:**
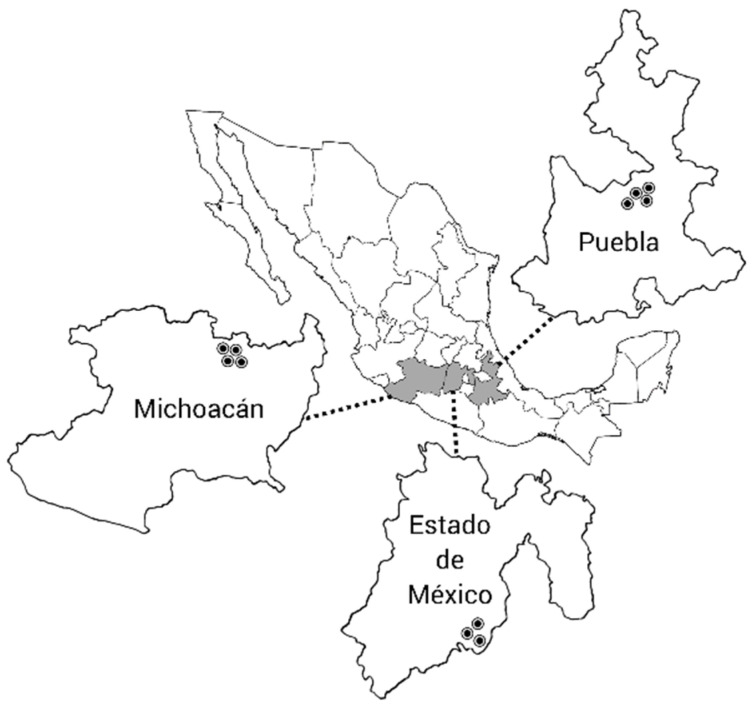
Region of Mexico were crops and sampling sites were developed.

**Figure 2 insects-12-00331-f002:**
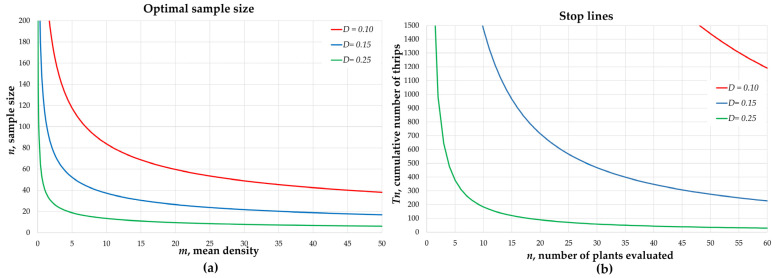
Sequential sampling with three levels of precision. (**a**) Relationship between the optimal sample size and the average density and (**b**) sampling stop lines based on accumulated counts.

**Figure 3 insects-12-00331-f003:**
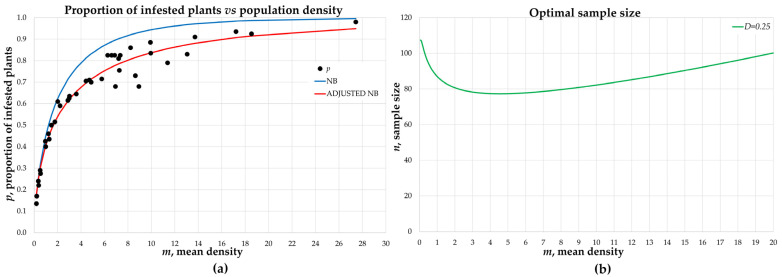
Binomial sampling. (**a**) Relationship between the proportion of infested plants and population density and (**b**) sample size as a function of the average density.

**Table 1 insects-12-00331-t001:** Statistical results for 1000 repetitions of bootstrap resampling. Sequential enumerative sampling with fixed precision (*D* = 0.10, 0.15 and 0.25).

Fixed Precision	Plot	*m* ± SE	*m′*	Sample Size			*D′* ± SE
				Mean	Maximum	Minimum
*D* = 0.10	1	0.37 ± 0.059	0.37	440.34	797	285	0.086 ± 0.002
	2	0.98 ± 0.112	0.97	268.88	426	186	0.077 ± 0.001
	3	1.80 ± 0.186	1.79	197.27	300	144	0.083 ± 0.002
	4	2.97 ± 0.266	2.98	153.14	216	109	0.077 ± 0.001
	5	5.82 ± 0.416	5.81	110.67	143	89	0.079 ± 0.002
	6	7.13 ± 0.491	7.11	99.50	124	82	0.082 ± 0.001
	7	9.82 ± 0.747	9.84	85.34	107	68	0.093 ± 0.002
	8	13.61 ± 0.907	13.62	72.42	91	60	0.091 ± 0.002
	9	18.37 ± 1.153	18.31	62.84	76	52	0.087 ± 0.002
	10	26.97 ± 1.592	27.15	51.99	65	44	0.097 ± 0.002
*D* = 0.15	1	0.37 ± 0.059	0.37	196.08	395	124	0.130 ± 0.002
	2	0.98 ± 0.112	0.98	118.19	189	82	0.123 ± 0.002
	3	1.80 ± 0.186	1.78	88.06	143	65	0.125 ± 0.002
	4	2.97 ± 0.266	2.98	68.48	97	52	0.118 ± 0.002
	5	5.82 ± 0.416	5.82	49.07	61	39	0.113 ± 0.002
	6	7.13 ± 0.491	7.09	44.33	59	37	0.117 ± 0.002
	7	9.82 ± 0.747	9.81	37.92	49	31	0.137 ± 0.003
	8	13.61 ± 0.907	13.65	32.29	40	26	0.135 ± 0.003
	9	18.37 ± 1.153	18.31	27.91	34	23	0.134 ± 0.003
	10	26.97 ± 1.592	26.93	23.03	29	20	0.137 ± 0.003
*D* = 0.25	1	0.37 ± 0.059	0.37	69.65	164	47	0.214 ± 0.005
	2	0.98 ± 0.112	0.98	42.56	66	31	0.198 ± 0.004
	3	1.80 ± 0.186	1.80	31.38	51	23	0.211 ± 0.005
	4	2.97 ± 0.266	3.01	24.63	40	20	0.206 ± 0.004
	5	5.82 ± 0.416	5.80	17.62	23	14	0.191 ± 0.004
	6	7.13 ± 0.491	7.05	15.98	21	13	0.194 ± 0.004
	7	9.82 ± 0.747	9.90	13.67	17	11	0.226 ± 0.005
	8	13.61 ± 0.907	13.52	11.64	14	10	0.218 ± 0.005
	9	18.37 ± 1.153	18.33	10.03	12	8	0.226 ± 0.005
	10	26.97 ± 1.592	26.81	8.31	10	7	0.224 ± 0.005

**Table 2 insects-12-00331-t002:** Statistical results for 1000 bootstrap resampling repetitions. Binomial sampling with fixed precision *D* = 0.25.

Fixed Precision	Plot	*m* ± SE	*m′*	Sample Size			*D′* ± SE
				Mean	Maximum	Minimum	
*D* = 0.25	1	0.37 ± 0.059	0.28	116.05	366	90	0.31 ± 0.005
	2	0.98 ± 0.112	1.11	121.29	193	104	0.24 ± 0.007
	3	1.80 ± 0.186	1.82	131.79	243	110	0.17 ± 0.004
	4	2.97 ± 0.266	3.03	150.14	286	109	0.16 ± 0.004
	5	5.82 ± 0.416	5.03	165.75	419	106	0.19 ± 0.003
	6	7.13 ± 0.491	6.02	176.65	487	108	0.20 ± 0.003
	7	9.82 ± 0.747	12.31	327.87	1023	118	0.26 ± 0.009
	8	13.61 ± 0.907	16.23	393.46	1644	113	0.21 ± 0.005
	9	18.37 ± 1.153	17.99	384.77	1416	124	0.13 ± 0.004
	10	26.97 ± 1.592	30.25	576.04	1336	154	0.16 ± 0.004

## Data Availability

The data presented in this study are available on request from the corresponding author. The data are not publicly available due to the authors would like to know how the requested data will be used.

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
