# Peer review of "Sequential and Binomial Sampling Plans to Estimate Thrips tabaci Population Density on Onion"

_insects, 2021, doi:10.3390/insects12040331_

Round 1

Reviewer 1 Report

The study by Soto-Rojas et al developed a sequential and bionomial sampling plans for estimating Onion thrips population density in order to safe effort and time without compromising precision.

The manuscript is generally well written and presented and is worthy of publication in Insects. However, I wish the authors could be more details in the method section e.g., a visual map of the sample sites used in this study will add to the clarity of this manuscript. How many people were involved in the actual Thrips sampling and how was human error minimized. Authors should provide  supplementary data table of the list of commercial sites sample in this study and include details such onion variety, sample date and season, stage of plant at sampling.

A general mixed model should also be used to verify that these variables did not skew or affect thrips density and distribution to validate the approach used in this study. Assuming homogeneity among the sampling sites/units without a statistical proof is not appropriate.

Author Response

Reviewer 1.

  • Add a visual map of the sampling sites used in this study to contribute to the clarity of the manuscript.

Response: Following the reviewer recommendation, a map of the sampling sites was added (Figure 1, Pag 3).

  • How many people were involved in the actual Thrips sampling and how was human error minimized?

Response: Sampling was done by three entomologists; all used a magnifying glass 10X and standardized their criteria by counting thrips on 10 plants and comparing their total numbers. The exercise was repeated five times. This explanation was included in the manuscript (L. 86-89).

  • Authors should provide supplementary data table of the list of commercial sites sample in this study and include details such onion variety, sample date and season, stage of plant at sampling.

Response: A file of supplementary material was included containing the information suggested by the reviewer.

  • A general mixed model should also be used to verify that these variables did not skew or affect thrips density and distribution to validate the approach used in this study. Assuming homogeneity among the sampling sites/units without a statistical proof is not appropriate.

Response: We appreciate all comments and this one in particular. An analysis was carried out and included to determine whether there was any effect of these variables. The variables did not have a significant effect on population density or distribution of the pest (L. 146-149).

The data were used to calculate Taylor parameters; and linear regression evidenced that Taylor's Power Law accurately relates the population mean and variance, even though environmental conditions were diverse (variety, plant phenology and sampling season).

Reviewer 2 Report

The authors studied the “Sequential and binomial sampling plans to estimate Thrips tabaci population density on onion”. While I have no objection against publishing the data, I have some issues that need addressing. The authors did not explain statistical methods in each result obtained in the experiments. Also, the manuscript suffers from poor English style and syntax errors. Writing should be improved by authors and additionally the article needs to be proofread by an English native speaker. The manuscript is in need of revisions before it is acceptable for publication. Please see my specific comments below:

Ls.10-14: Please revise these sentences to correct grammatical errors.

Ls.23-27: Summarize and revise these sentences to correct grammatical errors.

L.38: Keywords should be in alphabetic order. Also, keywords serve to widen the opportunity to be retrieved from a database. To put words that already are into title and abstracts makes KW not useful. Please choose terms that are neither in the title nor in abstract.

Ls.40-72: In introduction section, please revise this section to correct grammatical errors.

Ls.44-47: Confuse, rewrite.

L.49: Change “crop” by “plant”

Ls.70-72: Please revise this sentence to correct grammatical errors and eliminate wordiness.

Ls.81-82: Which developmental stages? Explain.

Ls.88-119: This subsection should be summarized.

Ls.126-128: Sentence is redundant, delete.

Ls.129-131: Please, revise this sentence to eliminate redundancy.

L.130:… (F1.39 = 1612, p <0.0001)? In materials and methods section, you did not explain the statistical method used in this study. I assume you used an analysis of variance. Check.

Ls.135-137: Please, revise this sentence to eliminate redundancy.

Ls.144-145: …such as that described by Torres 144 et al. [38]… This information should be in the discussion section.

Ls.152-155: Delete these sentences or place by materials and methods section

Ls.161-162: Again, delete this sentence.

Ls.184-186: Check the statistical method used in this study.

Ls.216-221: What is the purpose of this information?

Ls.223-288: Please revise this section to correct grammatical errors.

Author Response

Reviewer 2.

  • The authors did not explain statistical methods in each result obtained in the experiments.

Response: Considering the reviewer's suggestion, the statistical methods used were specified (L. 105-106, 136-140).

  • The manuscript suffers from poor English style and syntax errors. Writing should be improved by authors and additionally the article needs to be proofread by an English native speaker.

Response: The style and grammar were reviewed by a professional English native speaker. We hope that the manuscript now meets the standard English for scientific publications.

  • 10-14: Please revise these sentences to correct grammatical errors.

Response: The sentences were reviewed and corrected.

  • 23-27: Summarize and revise these sentences to correct grammatical errors.

Response: The sentences were reviewed and corrected.

  • 38: Keywords should be in alphabetic order. Also, keywords serve to widen the opportunity to be retrieved from a database. To put words that already are into title and abstracts makes KW not useful. Please choose terms that are neither in the title nor in abstract.

Response: Some keywords were added and sorted alphabetically.

  • 40-72: In introduction section, please revise this section to correct grammatical errors.

Response: The style and grammar was reviewed by an English native speaker.

  • 44-47: Confusing, rewrite.

Response: The sentences were modified.

  • 49: Change “crop” to “plant”
  • Response: Modified, as suggested by the reviewer.
  • 70-72: Please revise this sentence to correct grammatical errors and eliminate wordiness.

Response: The style and grammar were reviewed by a professional English native speaker. Unnecessary words have been eliminated.

  • 81-82: Which developmental stages? Explain.

Response: total number of thrips (except eggs) were counted on each plant. The explanation was included in the manuscript (L. 85-86).

  • 88-119: This subsection should be summarized.

Response: The subsection was reviewed. Some unnecessary information was eliminated.

  • 126-128: Sentence is redundant, delete.

Response: Redundant sentences were eliminated.

  • 129-131: Please, revise this sentence to eliminate redundancy.

Response: Redundant sentences were eliminated.

  • 130:… (F1.39 = 1612, p <0.0001)? In materials and methods section, you did not explain the statistical method used in this study. I assume you used an analysis of variance. Check.

Response: Considering the reviewer's suggestion, the statistical methods used were specified (L. 105-106, 136-140).

  • 135-137: Please, revise this sentence to eliminate redundancy.

Response: Redundant sentences were eliminated.

  • 144-145: …such as that described by Torres 144 et al. [38]… This information should be in the discussion section.

Response: The sentence and bibliographic citation was removed.

  • 152-155: Delete these sentences or place by materials and methods section

Response: Sentences were modified and placed in the Materials and methods section (L. 113-115)

  • 161-162: Again, delete this sentence.

Response: Redundant sentences were eliminated.

  • 184-186: Check the statistical method used in this study.

Response: Considering the reviewer's suggestion, the statistical methods used were reviewed. These methods have been used in similar studies.

  • 216-221: What is the purpose of this information?

Response: The sentences were modified and placed in the Discussion section (L. 240-243)

  • 223-288: Please revise this section to correct grammatical errors.

Response: The style and grammar were reviewed by a professional English native speaker.

Reviewer 3 Report

The manuscript by Soto-Rojas et al. investigates whether sequential and binomial samplings provide accurate estimates of thrips populations in onion crops. The authors find that both methods are accurate and reduce sampling time by 80%; however, they report that the accuracy of the sequential sampling is higher.

In general, the manuscript is well written and the information is important for better sampling thrips in onion crops. I, however, have a few comments and edits that hopefully will help improve the quality of the manuscript.

  1. The authors claim that they are testing sequential and binomial sampling as alternative methods to “conventional” sampling (lines 69-70). However, it is unclear to me what this conventional sampling entails. What do growers currently do to sample thrips in onions that is time consuming?
  2. If the goal is to determine more accurate and less time-consuming sampling methods like sequential and binomial sampling (as stated in line 71), I wonder why the authors did not include the standard “conventional” sampling in their study for comparison?
  3. I think some details are missing in the sampling techniques. First, I am unclear what the authors mean by “systemic sampling with random start” (lines 80-81). This needs to be explained. Also, I am assuming the sample size (200) means 200 plants sampled per plot, correct? Please specify. Other questions: where all samples taken from the plots in a single date? If so the authors need to indicate when the samples were taken (months) for the 41 plots and each year. The authors indicate that samples were taken “During 2015 and 2018”. I am assuming that they only collected from 2 years, correct? If not, clarify. This was confusing to me because the authors in line 234 indicate “from the first years” but there were only two years. Again, please clarify.

Specific edits:

Line 16: I think it should be “More than 40 onion plots” since there were only 41.

Line 17: provide author name after scientific name.

Line 23: provide author name after scientific name.

Line 23: change to “that causes economic”

Line 30: should be “2.586 and b” instead of “y”

Line 43: what does “t” means? “tons”? If so, please change.

Line 48: spell out genus at beginning of sentence.

Line 157: I think should be “using the figure”

Line 161: “reach the fixed”

Line 161: “in plot one, a”

Line 165: “in plot five, a”

Line 199: the sentence “>0.5 plant individuals per plant” does not make sense. Should it be “<0.5 individuals per plant”. Please check.

Line 202: “In plot one, a”

Line 228: I think T. palmi is mentioned here for the first time. If so, please provide full scientific name and name of author.

Line 270: provide name of author after scientific name.

Please check the “References” section. I do not think the words in the body of the titles need to be in upper case. Please check the author guidelines.

Author Response

Reviewer 3.

  • The authors claim that they are testing sequential and binomial sampling as alternative methods to “conventional” sampling (lines 69-70). However, it is unclear to me what this conventional sampling entails. What do growers currently do to sample thrips in onions that is time consuming?

Response: Generally, growers use non-probability sampling methods, which has no statistical basis, and their precision cannot be calculated. Agronomists with technical training use probability sampling methods (simple random sampling, systematic sampling or stratified sampling); these are the methods that we considered conventional; the main disadvantage is that their implementation is time-consuming. A sentence was included to describe the conventional sampling (L. 69-71.)

  • If the goal is to determine more accurate and less time-consuming sampling methods like sequential and binomial sampling (as stated in line 71), I wonder why the authors did not include the standard “conventional” sampling in their study for comparison?

Response: The comparison was made with systematic sampling, which is the one most commonly used by agricultural technicians. (L. 84-85.)

  • I am unclear what the authors mean by “systemic sampling with random start” (lines 80-81).

Response: In each plot, systematic sampling was carried out (sample size n = 200), plants were selected according to a random starting point using a fixed Interval. The sentence was modified in the manuscript (L. 84-85.).

  • I am assuming the sample size (200) means 200 plants sampled per plot, correct? Please specify.

Response: Thanks for the observation. The sample size was 200 plants sampled per plot. The sentence was corrected (L. 84-85.).

  • were all samples taken from the plots in a single date? If so, the authors need to indicate when the samples were taken (months) for the 41 plots and each year.

Response: In order to offer all this valuable information supplementary material was included.

  • The authors indicate that samples were taken “During 2015 and 2018”. I am assuming that they only collected from 2 years, correct? If not, clarify. This was confusing to me because the authors in line 234 indicate “from the first years” but there were only two years. Again, please clarify.

Response: We thank the reviewer for this observation. In fact, 41 samplings were performed in the period from 2015 to 2018 and an additional 10 samplings in the period 2019 to 2020. This explanation was included in the manuscript (L. 89-91)

  • Line 16: I think it should be “More than 40 onion plots” since there were only 41.

Response: Fifty-one samplings were carried out in 2015 to 2020. We revised the explanation and hope that it is now easier to understand (L. 89-91)

  • Line 17: provide author name after scientific name.

Response: Done.

  • Line 23: provide author name after scientific name.

Response: Done.

  • Line 23: change to “that causes economic”.

Response: Done.

  • Line 30: should be “2.586 and b” instead of “y”

Response: Done.

  • Line 43: what does “t” means? “tons”? If so, please change.

Response: Corrected.

  • Line 48: spell out genus at beginning of sentence.

Response: Corrected.

  • Line 157: I think should be “using the figure”

Response: Done.

  • Line 161: “reach the fixed”

Response: Corrected.

  • Line 161: “in plot one, a”

Response: Corrected.

  • Line 165: “in plot five, a”

Response: Corrected.

  • Line 199: the sentence “>0.5 plant individuals per plant” does not make sense. Should it be “<0.5 individuals per plant”. Please check.

Response: Thank you so much for the observation. Corrected.

  • Line 202: “In plot one, a”

Response: Corrected.

  • Line 228: I think T. palmi is mentioned here for the first time. If so, please provide full scientific name and name of author.

Response: Corrected.

  • Line 270: provide name of author after scientific name.

Response: Done.

  • Please check the “References” section. I do not think the words in the body of the titles need to be in upper case. Please check the author guidelines.

Response: Corrected.

Round 2

Reviewer 2 Report

The manuscript “Sequential and binomial sampling plans to estimate Thrips tabaci population density on onion” has been improved and all my questions were taken into account.
I recommend the publication in “Insects”.